# Recent Advances in the Lipid Nanoparticle-Mediated Delivery of mRNA Vaccines

**DOI:** 10.3390/vaccines11030658

**Published:** 2023-03-14

**Authors:** K. Swetha, Niranjan G. Kotla, Lakshmi Tunki, Arya Jayaraj, Suresh K. Bhargava, Haitao Hu, Srinivasa Reddy Bonam, Rajendra Kurapati

**Affiliations:** 1School of Chemistry, Indian Institute of Science Education and Research, Thiruvananthapuram 695551, India; 2Saveetha School of Engineering, Saveetha Institute of Medical and Technical Sciences, Saveetha University, Chennai 602105, India; 3Department of Applied Biology, CSIR-Indian Institute of Chemical Technology, Hyderabad 500007, India; 4Centre for Advanced Materials and Industrial Chemistry (CAMIC), School of Science, RMIT University, Melbourne, VIC 3001, Australia; 5Department of Microbiology and Immunology, University of Texas Medical Branch, 301 University Blvd, Galveston, TX 77555, USA; 6Institute for Human Infections & Immunity, Sealy Institute for Vaccine Sciences, University of Texas Medical Branch, Galveston, TX 77555, USA

**Keywords:** vaccine, mRNA, COVID-19, lipid nanoparticles

## Abstract

Lipid nanoparticles (LNPs) have recently emerged as one of the most advanced technologies for the highly efficient in vivo delivery of exogenous mRNA, particularly for COVID-19 vaccine delivery. LNPs comprise four different lipids: ionizable lipids, helper or neutral lipids, cholesterol, and lipids attached to polyethylene glycol (PEG). In this review, we present recent the advances and insights for the design of LNPs, as well as their composition and properties, with a subsequent discussion on the development of COVID-19 vaccines. In particular, as ionizable lipids are the most critical drivers for complexing the mRNA and in vivo delivery, the role of ionizable lipids in mRNA vaccines is discussed in detail. Furthermore, the use of LNPs as effective delivery vehicles for vaccination, genome editing, and protein replacement therapy is explained. Finally, expert opinion on LNPs for mRNA vaccines is discussed, which may address future challenges in developing mRNA vaccines using highly efficient LNPs based on a novel set of ionizable lipids. Developing highly efficient mRNA delivery systems for vaccines with improved safety against some severe acute respiratory syndrome coronavirus 2 (SARS-CoV-2) variants remains difficult.

## 1. Introduction

Advances in messenger RNA (mRNA)-based delivery technologies have shown therapeutic potential in various biomedical applications, including protein replacement therapies, vaccines, cellular reprogramming, and cancer immunotherapies [1,2,3]. To achieve optimal therapeutic benefits, mRNA molecules must be protected from degradation and delivered to specific target cells to produce the desired proteins. However, targetability, stability, and endosomal escape remain major challenges for mRNA delivery systems, highlighting the importance of safe and effective mRNA delivery [4]. Numerous biocompatible, biodegradable lipids, polymers, and protein-based derivatives have been employed in creating mRNA-based therapies [5].

LNPs, in particular, have emerged as a well-studied class of vehicles in the biopharmaceutical industry as prospective delivery systems for various nucleic acid therapies, including oligonucleotides. LNPs have several advantages over viral vectors for gene therapy applications, including moderate or less immunogenicity, a large payload, simple production, and great scalability. Furthermore, LNPs are designed for specific cell targets and disease-related applications by adjusting the lipid-to-mRNA ratio. The use of LNPs in COVID-19 mRNA vaccines is gaining attraction, since they are crucial for preserving and delivering the payload (mRNA) to specifically targeted cells [4]. In particular, mRNA-LNP vaccines against COVID-19 are currently in clinical usage, marking a novel strategy for mRNA-based treatments [6,7].

Insights gained over many years into LNPs have considerably aided the meteoric rise of RNA therapeutics in recent years. Onpattro^®^ (an RNA interference drug that was the first FDA-approved LNP-nucleic acid therapeutic for the treatment of polyneuropathy) [8] and COVID-19 (coronavirus disease 2019) vaccines (Pfizer-BioNTech and Moderna mRNA-LNPs) are just a couple examples of how RNAs can be delivered to target cells by clinically translatable LNPs [9]. Scientists spent over a decade researching and evaluating various ionizable lipids and LNP-based formulations, until Onpattro^®^ was certified in 2018. Clinical development of other LNP-based RNA therapeutics, especially mRNA vaccines (including two recent breakthrough LNP-based mRNA vaccines, mRNA-1273 (Moderna), and BNT162b2 (BioNTech) for COVID-19), has been greatly accelerated in response to the success of MC3 (ionizable lipid used for Onpattro^®^ product by Alnylam^®^ Pharmaceuticals) [10,11]. However, more effective vaccinations against some recently developing severe acute respiratory syndrome coronavirus 2 (SARS-CoV-2) mutations remains a formidable obstacle to mRNA-based vaccine development [7].

Focusing on their pharmacological applications, we summarize the current understanding of LNP design composition and features. We then talk about how ionizable lipids function during mRNA transport. We also provide a panorama of traditional techniques compared to the significance of modern microfluidics technology in preparing LNPs. Finally, we discuss the effects of LNP pharmacokinetics on immunogenicity and provide a glimpse into the future of LNPs and mRNA therapies.

## 2. The Structural Composition of LNPs and Their Role in Developing mRNA Therapeutics

LNPs are generally spherical systems with at least one lipid bilayer and an aqueous interior compartment, which provide several benefits, including formulation simplicity, self-assembly, biocompatibility, high bioavailability, and the potential to transport large payloads [12]. These reasons make LNPs the recommended type of formulation for nanomedicines approved by the FDA [13]. LNPs have been extensively studied and employed successfully in clinics for the delivery of numerous macromolecules and small molecule medicines, including nucleic acids.

Unlike classical bilayered liposomes, LNPs have a complex internal architecture with enhanced physical stability, due to their structural properties. Size, surface charge, and ligand surface functionalization are key physicochemical features of LNPs that can be tuned during synthesis to improve in vitro and in vivo stability and facilitate unique nucleic acid delivery capabilities [14,15]. Upon administration, LNPs are internalized by host cells; thereby, the encapsulated mRNA cargo will be delivered inside the cytosol, and the mRNA sequences are further translated into targeted protein by the ribosome of host cells (discussed in detail in the following sections) [16]. One of the key obstacles of mRNA vaccines is that naked mRNA is quickly degraded upon administration by enzymes ribonucleases (RNase), reducing intracellular stability and further minimizing translation efficiency. Despite mRNA’s extensive therapeutic utility in the pharmaceutical industry, intracellular targetability remains challenging due to its high molecular weight, polyanionic nature, and intrinsic chemical instability. For this reason, LNPs and other lipid-based nanomedicine systems are crucial for the safe and efficient delivery of exogenous mRNA to the targeted cells.

### 2.1. Material Aspects and Structural Design of LNPs for mRNA Therapeutics

The four lipid components listed below (for more information, see Figure 1 and Table 1) are the most common building blocks for LNPs, including (i) ionizable lipids, which are primarily required for mRNA complexing; (ii) helper lipids, which enhance the properties of LNPs (stability and delivery efficiency); (iii) cholesterol, which provides structural stability to the LNPs (vehicle); and (iv) surface PEGylation with poly(ethylene glycol) (PEG) or PEG-derivatives, which reduces host-immune recognition and improves systemic circulation [9,17,18]. Among all these lipids, ionizable lipids are the essential components of most LNPs, acting as a primary driver to deliver mRNA through high mRNA encapsulation, enhancing stability, and pH-sensitive in vivo delivery [18]. Each component of LNPs and its function in mRNA delivery will be described in detail.

#### 2.1.1. Ionizable Cationic Lipids

Ionizable lipids, such as OF-Deg-Lin and FTT5 (Table 1), are protonated at low pH, making them positively charged inside the cell and uncharged/neutral at normal pH in the bloodstream [18,19]. These positively charged lipids stabilize the nucleic acid and prevent degradation by nucleases [6]. Ionizable lipids contribute to cellular uptake of the LNPs and endosomal escape by acquiring the positive charge in the acidic environment of the endosome (discussed in detail in the following sections) [18,20].

#### 2.1.2. Neutral/Helper Phospholipids

In addition to ionizable lipids, LNPs also comprise helper lipids, such as phospholipids (e.g., phosphatidylcholine and phosphatidylethanolamine, 1,2-distearoyl-sn-glycero-3-phosphocholine (DSPC) and 1,2-dioleoyl-sn-glycero-3-phosphoethanolamine (DOPE), respectively). These phospholipids improve the particle stability, membrane integrity, and delivery efficacy of LNPs [4]. These lipids (DOPE and DSPC) improve transfection efficiency by promoting membrane fusion while stabilizing the LNPs by generating various geometries with acyl chains [6]. They destabilize the endosomal membrane and promote the endosomal escape of LNPs; the ratios of different lipid composition(s) influence the effectiveness of the nanoparticles. LNPs with higher amounts of helper lipids are used for mRNA delivery. However, mRNA delivery is enhanced in DOPE than in DSPC, as it can undergo a conformational change from a stable lamellar phase to an unstable hexagonal phase, resulting in membrane fusion. In contrast, the phosphocholine-containing lipids inhibit membrane fusion-mediated endosomal escape [4,21].

#### 2.1.3. Cholesterol

Recent research has demonstrated that the hydrophobic and rigid lipid cholesterol can fill in the spaces between other lipids in the vesicle membrane and potentially change the stiffness and integrity of the membrane, increasing the stability of the particles [4,6,22]. Additionally, incorporating the cholesterol showed improved delivery efficacy, increased the circulation half-life of nanoparticles, and enhanced the transfection efficiency of LNPs by promoting membrane fusion and endosomal release [4,13,23]. For example, Patel et al. demonstrated that incorporating cholesterol analogues with C-24 alkyl phytosterols into LNPs (eLNPs) could enhance gene transfection due to higher cellular uptake and retention followed by a steady release of encapsulated mRNA [24]. An increase in cholesterol percentage lowers the transitional temperature of the membranes of LNPs [24]. Moreover, the biocompatibility of the LNPs is enhanced as cholesterol is one of the major components of biological membranes [25].

#### 2.1.4. Lipid Anchored Polyethylene Glycol (PEG) Constructs

PEG-lipids constructs, which are PEG molecules connected to alkyl chains, operate as an anchor within the LNP layer. Including PEG lipids in LNPs can prevent opsonization by serum proteins and reticuloendothelial clearance, both of which are crucial for enhancing biodistribution [26,27]. In addition, incorporating PEG-lipid into LNPs can affect the size and surface charge of the nanoparticles and improve the surface properties by restricting access through steric hindrance to the surface [6,28]. PEG-lipids can further prevent the blood plasma protein surface adsorption, thereby increasing the circulation lifetime in the bloodstream and minimizing aggregation behavior [4]. Furthermore, the PEG-lipid composite coating of LNPs facilitates receptor-mediated cellular absorption by exchanging with serum proteins, such as ApoE [22,24]. Additionally, PEG-lipids can prevent endosomal escape by blocking the interaction between liposome and endosomal membrane sterically and electrostatically [20,22,29,30].

**Table 1 vaccines-11-00658-t001:** Shows the chemical structures of different lipids used in LNPs for mRNA delivery.

Lipid	Function	Ref
Cationic ionizable lipids
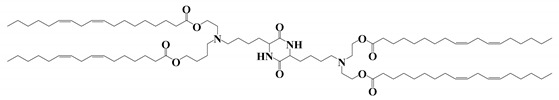 OF-Deg-Lin	Selective delivery of mRNA into B lymphocytes	[31]
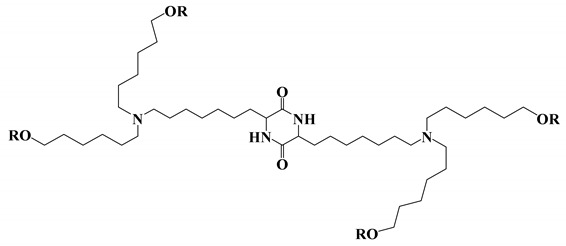 OF-C4-Deg-Lin	Selective delivery of siRNAs and mRNAs	[32]
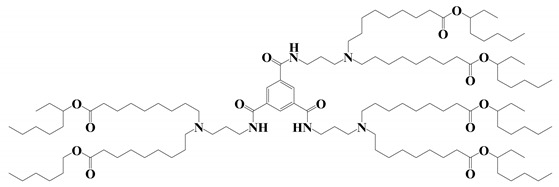 FTT5	In vivo delivery of mRNA encoding human factor VIII and base editing components	[33]
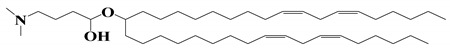 Dlin-MC3-DMA	Used in albumin receptor-mediated delivery of mRNA to the liver	[34]
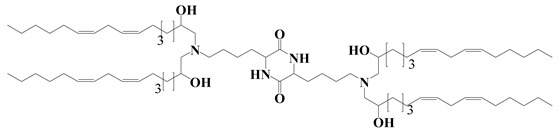 OF-02	Enhanced hepatic mRNA delivery	[35]
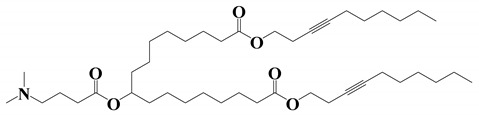 A6	Albumin receptor mediated mRNA delivery	[34]
Neutral/helper lipids
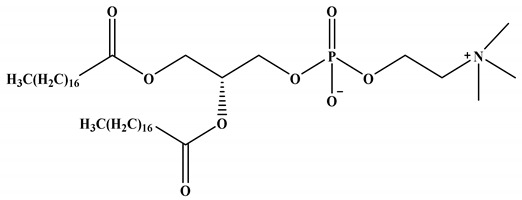 DSPC	Used in mRNA vaccines and vaccine candidates, including COVID-19.	[6]
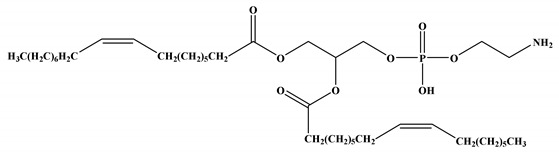 DOPE	Delivery of a variety of nucleic acids.	[25]
PEG lipids
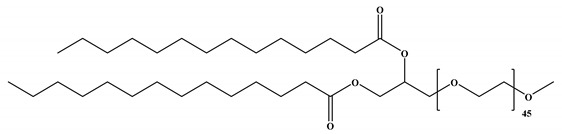 PEG2000-DMG	Used in mRNA vaccines and vaccine candidates, including COVID-19.	[6]
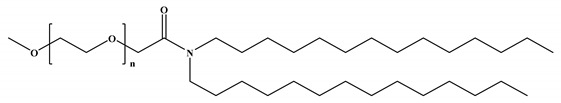 ALC-0159	Delivery of mRNA vaccines.	[6]

### 2.2. Role of Ionizable Lipids in mRNA Delivery

Among the four types of lipids, ionizable lipids are the most important lipids complex with the negatively charged mRNA and are responsible for endosomal escape of mRNA subsequently releasing in the cytosol, thanks to the unique properties of ionizable lipids and their pH-dependent surface charge. Notably, these lipids are neutral in the physiological pH environment but positively charged under low pH [17,36]. Hence, at the physiological pH, the neutral lipids have the least interaction with the anionic cell membrane, which enhances the biocompatibility of the LNPs. At the same time, inside the endosome (acidic pH), the cationic ionizable lipids interact with the anionic endosomal phospholipids to form cone-shaped ion pairs known as the inverted hexagonal II (HII) phase. These structures distort the bilayer of LNPs and facilitate membrane disruption and payload release into the cytosol [4]. According to the literature, ionizable lipids are broadly categorized into five types, as shown in Figure 2: (i) unsaturated, (ii) multi-tail, (iii) polymeric, (iv) biodegradable, and (v) branched tail ionizable lipids [36].

Unsaturated ionizable lipids constructs have increased cis double bonds in the tail, making the bilayer structure unstable and helping to form inverted hexagonal structures [37] efficiently. As a result, it improves membrane destruction and, hence, allows for cargo release; such lipids are OF-02 and A6 [35,36]. As the name implies, multi-tail ionizable lipids have more than two tails (e.g., C12-200). Compared to two-tailed ionizable lipids, these multi-tail endings aid in forming non-bilayer, cone-shaped structures, which intensify endosomal destruction and promote highly effective cargo release [36,38]. On the other hand, polymeric ionizable lipids are synthesized by replacing the free amine groups in cationic polymers with alkyl tails [36]. These lipids stimulate the formation of cone-shaped structures through hydrophobic interaction. G0-C14, for example, is a polymeric ionizable lipid employed in synthesizing lipid nanoparticles and has applications in cancer RNA therapies [36,39]. However, the main disadvantage of polymeric ionizable lipids is that they increase the complexity of lipid nanoparticle formation, due to undesired substitution compounds, even after purification. Furthermore, the cytotoxicity is increased by the polycationic core and non-biodegradable polymeric structure [4].

The insertion of cleavable ester groups within the hydrophilic alkyl chains of the lipid structure is one method for developing biodegradable ionizable lipids [40,41,42]. These ester groups must be easily degraded in the cytosol, while remaining stable at physiological pH. For example, 304O13 (Figure 2) is a biodegradable lipid with comparable potency to its non-biodegradable equivalent, C12-200, and lower toxicity even at higher dosages [36,41]. As a result of fast hydrolysis, these lipids have a decreased potency of gene (protein) expression, compared to their non-biodegradable analogue. However, employing lipids containing secondary esters (i.e., constituted of an ethanolamine headgroup, a primary ester at the C8 position, and secondary esters in the second lipid tail) resulted in an optimum balance of in vivo protein expression and lipid clearance [36,43]. Increased tail branching can also improve lipid performance; for example, acrylate-based ionizable lipids performed better than methacrylate-tailed ones [44,45]. Compared to other lipids, these lipids had larger cationic charges at endosomal acidic pH, which improved endosomal escape, due to wider lipid tails and simpler cone structure formation. Furthermore, in the case of COVID-19 vaccines, the ionizable lipids employed to deliver the mRNA are branched-tail ionizable lipids [17]. FTT5 is a well-studied ionizable lipid with enhanced mRNA delivery for protein supplementing applications [36,43]. However, how these branched tails affect the quality of LNPs has been studied poorly, due to the lack of commercial availability of these branched tail components [17]. These examples highlight the significance of ionizable lipids the LNP-based therapeutics.

Chen K et al. 2022, recently published a novel set of unique ionizable lipids, known as 4N4T, to construct a new series of LNPs known as 4N4T-LNPs by squeezing the lipid organic phase with aqueous mRNA solution into a microfluidic chip [7]. Compared to the approved SM-102-LNPs, 4N4T-LNPs show higher mRNA translation efficiency against SARS-CoV-2 and its variants, including Delta and Omicron. Overall, this study found that 4N4T-based lipid delivery technology could aid in the development of sophisticated, efficacious mRNA therapeutics for infectious diseases [7].

Interestingly, Hashiba et al. 2022, reported that ionizable lipid branching influences stability, fusogenicity, and functional mRNA delivery. Branched lipids (CL4F 8-6) with a high level of symmetry contributed ideal features for efficient intracellular distribution and stable formulations, providing new insights into rational lipid design and successful gene therapy applications [17]. Riley et al. 2021, developed a library of ionizable LNPs for in utero mRNA delivery to mouse fetuses and demonstrated a prenatal LNP delivery platform for erythropoietin (EPO) mRNA to hepatocytes in the fetal circulation for protein replacement therapy [46].

## 3. Moving beyond Classical Lipid Carriers to Specialized LNPs Design

The classical liposomes, an early version of LNPs, are versatile nanocarriers that deliver both hydrophobic and hydrophilic therapeutic agents. In recent years, much attention has been paid to developing subsequent generations of LNPs, such as solid LNPs (SLNPs), nanostructured lipid carriers (NLCs), and cationic lipid-nucleic acid complexes with more structural complexity and greater physical stability of the delivery carrier [6,47]. Notably, LNPs took advantage of spontaneous formation or self-assembly, due to the interactions between the charged groups in the lipidic mixture and charged nucleic acids. Intermolecular interactions aid in the formation of spherical nanostructures from lipid components. In general, various methods have been used to formulate LNPs, including nanoprecipitation, thin-film hydration, extrusion, homogenization, microfluidic mixing, and others. Among the existing methods, the latest approach is microfluidic mixing, since it produces uniform, size-controlled, stable, and reproducible LNPs. In addition, recent advancements have shown microfluidic process is one of the most reliable methods, compared to conventional methods [14].

The development of more adaptable, highly effective, and biocompatible systems is made possible by ongoing efforts to synthesize and evaluate a variety of LNPs by chemically modifying their molecular architectures. The rapid development of mRNA vaccines was only possible with advances in screening the latest lipid constructs and LNP technologies to deliver nucleic acids. In particular, cationic and ionizable lipids are preferred because of their inherent tendency to self-assemble into LNPs with nucleic acids via intermolecular interactions, which will help efficiently deliver the payload [48].

### Microfluidics over Other Conventional Methods

Liposomes and LNP were produced using the standard lipid thin film hydration process. However, this method has various drawbacks, including heterogeneous particle size distribution caused by forming nanoparticles of varying particle sizes. Henceforth, a subsequent size tuning approach is necessary to formulate uniform, homogenous LNPs. Furthermore, post-processing methods, such as extrusion, sonication, and others, may result in morphological alterations to the final nanoparticles produced [49]. To address these limitations, researchers recently developed a microfluidics tool that simplified production processes [50].

Microfluidics tools have numerous benefits, including configurable parameters (flow rate, controlled mixing, temperature, and time), which aid in producing accurate LNPs to improve transfection efficiency [51,52]. Importantly, the flow of ingredients and the speed of mixing during the rapid production process control LNP production processes. Scale-up is also feasible with minimum time and cost, which adds a significant advantage in formulation development, especially in emerging situations, such as the COVID-19 pandemic, allowing facile clinical translation [53]. Numerous articles published over the past five years demonstrated the importance of the microfluidics approach for synthesizing LNPs with substantial benefits for precise and rapid manufacturing. Furthermore, the recently developed mRNA-based COVID-19 vaccines by Pfizer/BioNTech and Moderna are produced on a massive scale utilizing this microfluidic approach, emphasizing the technique’s importance in the field of vaccine production for use in emergency situations [54].

Although the predictable and rational design of LNPs for nucleic acid delivery to extrahepatic tissues remains difficult (LNPs resemble very-low-density lipoprotein and adsorb apolipoprotein E in blood plasma, resulting in preferential accumulation in the liver and uptake into hepatocytes), this challenge must be overcome in order to realize the full potential of mRNA LNP technology for broad therapeutic developments [55]. However, such mechanisms are effective in treating liver diseases, including cancer. A recent interesting new technology, termed as selective organ targeting (SORT), was developed, whereby nanoparticles are methodically tailored to deliver therapeutic compounds to both hepatic and extrahepatic regions (Figure 3) [56]. According to Wang et al. 2022, SORT LNPs are tunable SORT molecules that target the payload to the liver, lungs, and spleen of mice after intravenous delivery. The engineered SORT LNPs are able to target specific cells and organs in the body via passive, active, and endogenous targeting methods, each of which necessitates a different set of design requirements (Figure 3) [55]. The authors presented procedures for preparing 4A3-SC8- and MC3-based liver, lung, and spleen SORT LNPs utilizing three approaches (pipette mixing, vortex mixing, and microfluidic mixing) to strategies for small-, medium-, and large-scale manufacturing.

## 4. Applications of LNPs in mRNA Delivery

Before discussing the applications of LNPs, a brief discussion on mRNA and its therapeutic potential is placed here. Since its first isolation in 1961, mRNA (that encodes the protein of interest) research has taken several paths, which made us understand its diversified functions and modification-mediated potential for therapeutic applications [1,57]. As a result of the COVID-19 pandemic, nucleic acid therapeutics (NATs), particularly mRNA vaccines, potentials have been enabled for emerging infectious diseases. The translation of host genetic information (DNA) into proteins by ribosomes in the cytoplasm is mediated by mRNA (Figure 4).

There are several approaches for modifying mRNA sequence to improve mRNA-based vaccines [2].

By binding to eukaryotic translation initiation factor 4E, synthetic cap analogues and capping enzymes stabilize mRNA and speed up the translation of proteins.Multiple adenines in the Poly(A) tail stabilize mRNA and enhance the translation of proteins.Modified nucleosides and codon sequence optimization avoid the innate immune activation and increase the translation efficiency, respectively.The use of mRNA purifying methods, such as cellulose, RNase III, and fast protein liquid chromatography (FPLC), decreases the presence of double-stranded RNA.

### 4.1. Use of mRNA as Prophylactics and Therapeutics

Despite the usage of DNA vaccines, RNA-based vaccines have grown in popularity in the 21st century. Interestingly, mRNA vaccine technology purely depends on the genetic code of the virus; since the SARS-CoV-2 genetic sequence was published [58], virally generated self-amplifying RNA and non-replicating mRNA are being used abundantly for vaccine development. Indeed, during the COVID-19 pandemic, mRNA vaccines have become the only vaccine source available to combat the rapid spread of disease. While multiple pharmaceutical companies have investigated several vaccine technologies against COVID-19, the FDA has approved two vaccines (first for emergency use and then for full approval), Pfizer/BioNTech’s and Moderna’s Spikevax (Table 2) [59]. Both use the complete sequence of spike and receptor-binding domains of SARS-CoV-2 as target antigens. However, mRNA-based preclinical studies have been reported since 1990. Its poor stability as naked mRNA and tremendous capacity to induce an innate immune response (particularly double-stranded RNA) has halted the progress of RNA-based therapeutics. The emergence of advanced nanotechnologies for the protection and targeted delivery of water-soluble molecules have shined a light on developing the LNP-mRNA vaccines. Nucleic acid delivery to the cytosol, without hampering the cell wall integrity, and degradation by nucleases is a rate limiting step for targeted delivery. Despite mechanical and viral particle-mediated targeted delivery, chemical based targeted delivery via LNPs have been the ease of the process and have been less toxic, with less immunogenicity. During the start of the decade, several clinical trials have been initiated on mRNA therapeutics, which cover a range of different types of diseases, such as cancer, infectious diseases, protein replacement therapy, and others. Correspondingly, promising results have been shown in terms of disease alleviation, reduction of pathogenicity, and others. Interestingly, several diseases or disorders are targeted by using mRNA therapeutics. For the proof-of-concept studies, in vitro transcription (IVT)-produced mRNA is more suitable (Figure 5). To characterize the efficiency of mRNA in vitro, several transfection reagents are available to facilitate mRNA translation and targeted protein production, although they are not as efficient as targeted delivery systems. The Figure 5 shows the most successful mRNA vaccines. As explained earlier, a suitable (perfect mix) vehicle is needed for the mRNA vaccine delivery. LNPs have become the one of the most characterized mRNA delivery vehicles.

### 4.2. LNP Role in mRNA Delivery

Using lipid-encapsulated forms of sequence-optimized mRNA vaccines has shown strong immunity (both in preclinical and clinical) against infectious disease targets like the flu virus, Zika virus, rabies virus, and others, especially in the last few years [60]. Of note, the LNPs act as potential vaccine adjuvants [59], which helps in shaping antigen-specific immunity [61]. Despite several components of the LNP, ionizable lipids have a prominent role, including forming LNPs and the targeted delivery of mRNA into the cytosol. The ionizable lipid imparts the neutral charge to the LNPs at physiological pH and ionizes at lower pH in the endosome/lysosome, followed by the delivery of intact mRNA to the cytoplasm, where it can then be translated into the encoded protein. Despite mRNA, LNPs also induce the innate immune signaling pathways that aid in the adaptive immune responses. Similar to other potential adjuvants, LNPs (based on the composition) also induce the activation of several signaling pathways, including nuclear factor kappa B, interferons, inflammasome activation, and other [46].

In the future, it will be essential to compare and understand the immune pathways activated by different mRNA vaccine platforms, improve current approaches based on these mechanisms, and start new clinical trials against more disease targets.

### 4.3. Salient Features of LNP-Based Therapeutics [2,4,59]

Naked mRNA is unsuitable for therapeutic purposes, as it is rapidly degraded by extracellular RNases. Several nanotechnology platforms have been set and optimized for mRNA-targeted delivery.As mRNA is thermolabile, LNPs would enhance its stability and half-life at room temperature and be useful for avoiding vaccine cold chain.Modifications of the type of delivery system (carrier molecules) rule out the organ-specific mRNA delivery (for lung, spleen, liver, etc.) and its in vivo half-life.Composition of LNP could decide the type of immune response induced.LNPs act as adjuvants systems for mRNA vaccines.The stable LNPs make the lower dose of mRNA work.Composition of LNPs also decides the number of booster doses required if it is admixed with a suitable adjuvant.Moreover, for chronic treatments, multiple administration via different routes of administration is possible.

### 4.4. Mechanism of mRNA Vaccines

Even though the needle-free route of administration is long awaited [62,63,64], mRNA encapsulated by LNPs is administered mostly by the intramuscular route, and several study results support this route of administration [65,66,67]. Upon injection, local innate immune cells are recruited to the site of injection, followed by the uptake of LNPs by innate immune cells [68]. Once the LNPs reach the cytosol, lower pH enables the LNPs to release mRNA, followed by the translation of targeted antigens (Figure 5). The antigens are presented to the T helper cells (CD4^+^ T cells) and release several cytokines and chemokines, which further help in the shaping the antigen-specific immune responses, including B cell differentiation and plasma cell development [69]. The two successful vaccines (Pfizer/BioNTech and Moderna’s Spikevax) have confirmed this mechanism in both preclinical and clinical studies (Figure 6) [70].

**Table 2 vaccines-11-00658-t002:** FDA approved COVID-19 mRNA vaccines [71,72].

Name	mRNA Specific to	LNP Composition	Adverse Effects
BNT162b2	Spike glycoprotein of SARS-CoV-2	lipids ((4- hydroxybutyl)azanediyl) bis(hexane-6,1- diyl)bis(2- hexyldecanoate), 2 [(polyethylene glycol)- 2000]-N,Nditetradecylacetamide, 1,2-distearoyl-snglycero- 3-phosphocholine, and cholesterol), potassium chloride, monobasic potassium phosphate, sodium chloride, dibasic sodium phosphate dihydrate, and sucrose	Myocarditis Pericarditis
mRNA- 1273	Spike glycoprotein of SARS-CoV-2	LNP: Proprietary Ionic lipid SM-102, polyethylene glycol (PEG) 2000, dimyristoylglycerol (DMG), cholesterol, 1,2- distearoyl-sn-glycero-3- phosphocholine [DSPC]), tromethamine hydrochloride, acetic acid, sodium acetate, and sucrose	Myocarditis Pericarditis
LUNARCOv- 19 (ARCT- 021)	Self-replicating mRNA specific to Spike glycoprotein of SARS-CoV-2	Arcturus Therapeutics proprietary ionizable lipid, DSPC, cholesterol, and PEG2000-DMG dissolved in ethanol	NA

NA: not available.

## 5. Expert Opinion

Since its discovery, NATs (siRNA) started seeing the light in the twenty-first century with the approval of patisiran (2019) and mipomersen (2013). Even though NATs have been used since the 20th century, the COVID-19 pandemic witnessed its potential for public health in protecting from diseases. However, the delivery or targeted delivery of nucleic acid therapeutics is a major concern for maintaining it in stable condition until it reaches the targeted site to prevent off-target side effects.

Since the 1995 approval of the first liposomal drug delivery system, i.e., Doxil, several nanomedicine platforms for drug delivery have been emerged, such as micelles, lipid nanocapsules, solid LNPs lipid nanoparticles, and many other polymeric nanoparticles. Interestingly, few polymers (e.g., Poly (lactic-co-glycolic acid) (PLGA), Polyethylene-imine (PEI), chitosan, and others) that have been used to deliver the NATs, particularly DNA vaccines, have been shown significant results in both preclinical and clinical studies [73].

Nevertheless, LNPs succeeded in protecting NATs degradation and site-specific delivery. Therefore, NATs and LNPs have been the way out in targeting many diseases that do not have any acceptable, approved, and available medical alternatives.

Despite the efficacy of mRNA vaccines, the mutation rate of SARS-CoV-2 is alarming mRNA technology to focus more on the multi-antigenic vaccines, which could prevent the virus escape, induced by the low mutations, from the host immune response. Similarly, the use of potential vaccine adjuvants should also be considered, which not only enhances the antigen-specific immune response, but also helps with the dose, cost, scalability, and mass production of mRNA vaccines [61]. On the other hand, unlike DNA vaccines, mRNA vaccines always require ultra-cool temperatures or a cold chain cycle (−20–−80 °C) to deliver to the low-income countries. Despite mRNA vaccine efficacy, the adverse effects should not be overlooked. Even though lesser incidences of Myocarditis and pericarditis have been reported, the incidence rate in young adults in age range of 18–30 years is considerable [71]. Furthermore, either with booster doses or after bivalent immunization, there was a higher rate of inability to work, and the use of PRN (pro re nata) medications in healthcare workers are reported [72]. Therefore, focused research should be encouraged regarding the reasons for these adverse reactions.

## 6. Unanswered Questions

Are the ionizable lipids deciding the fate of immune response against mRNA-encoded protein? Changing the ionizable lipid composition will vary the type of immune response, B or T cell response. Perhaps some mRNA vaccines show T cell-mediated protection without having notable neutralizing antibody-inducing capacity [74].As the documentation of evidence showing the mRNA vaccine induced local and systemic adverse events [75], are these events related to the vaccine or the presence of minute amounts of dsRNA?Several studies confirm the adjuvant effect of LNPs in LNP-mRNA vaccines. Therefore, it is common to have some level of mild adverse events. Is this related to the lipid composition, e.g., anti-PEG antibodies, due to the presence of PEG [76,77]?Not only RNA-based impurities, but also lipid-based impurities, require structural elucidation in order to avoid loss of stability and activity. Do we have standard regulatory methods for measuring impurities? Several new strategies have been developed in codon optimization by keeping the mRNA template intact. Do these lead to transient immunogenicity (if not immunosilent), rather than durability, as they lack naïve uridine or modified uridines?Do we have enough data to support the best route of mRNA vaccine administration for durable immune responses with fewer boosters?Studies that found vaccine mRNA in human blood [78] and breast milk [79] raise questions about how stable they are in terms of biodistribution and pharmacokinetics. Is self-amplifying mRNA technology a possible way to make vaccines in the future?

## Figures and Tables

**Figure 1 vaccines-11-00658-f001:**
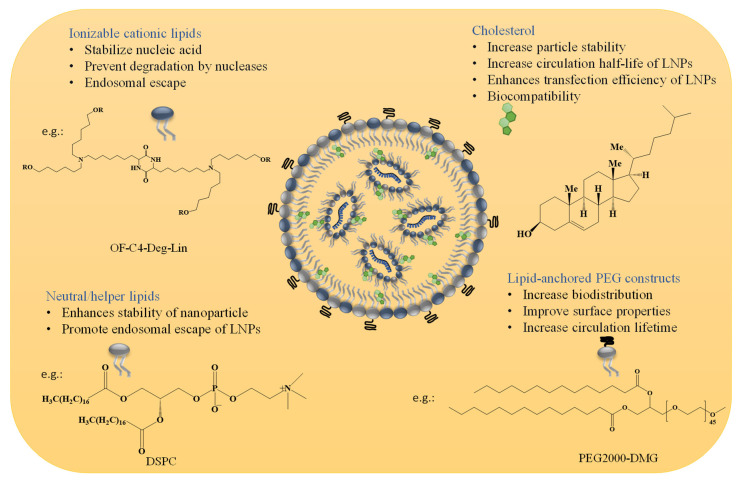
Composition of LNPs and importance of each component.

**Figure 2 vaccines-11-00658-f002:**
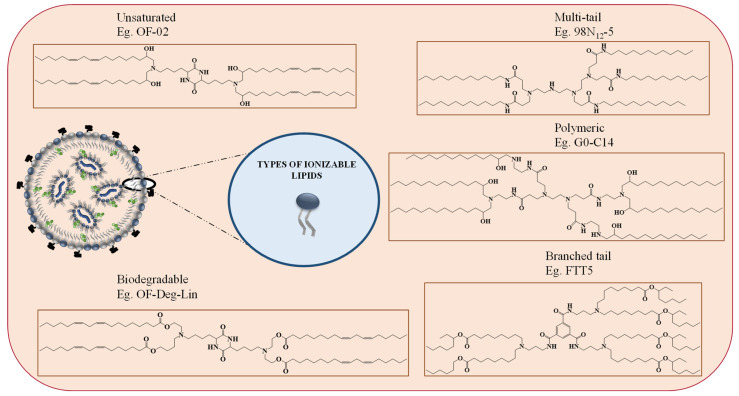
Shows the different types of ionizable lipids used in the composition of LNPs.

**Figure 3 vaccines-11-00658-f003:**
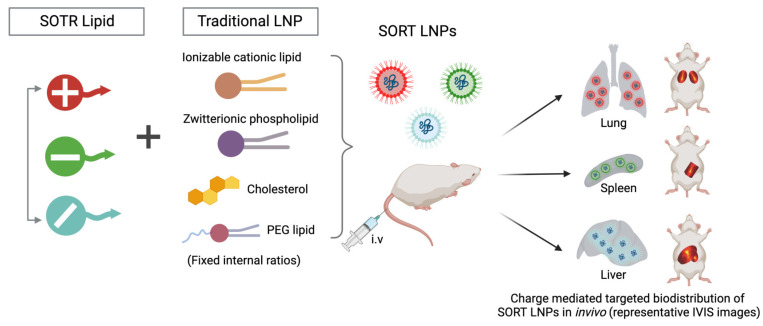
Shows the addition of a SORT lipid molecule to typical four-component LNPs alters the in vivo delivery profile of the resultant five-component SORT LNPs, allowing for tissue-specific distribution of mRNA to the liver, lungs, and spleen of mice following IV injections.

**Figure 4 vaccines-11-00658-f004:**
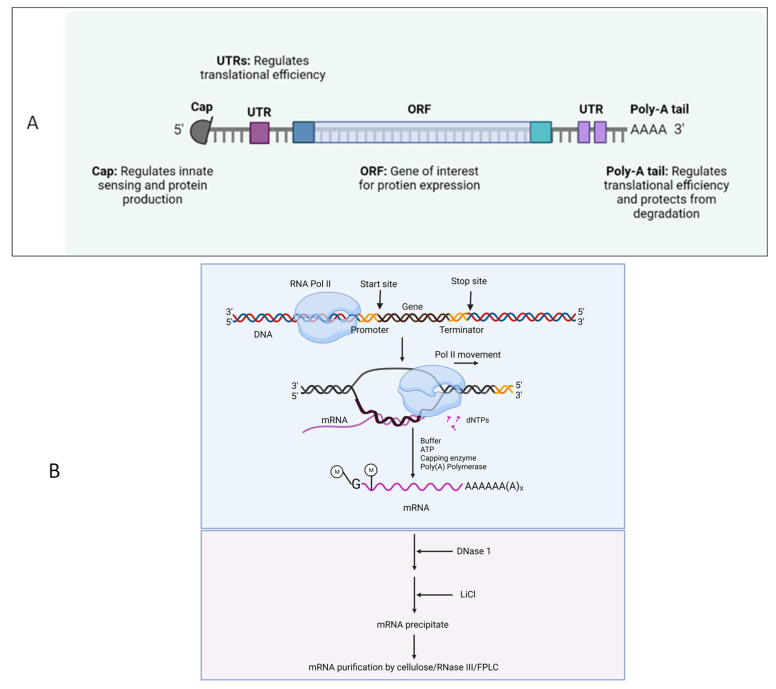
(**A**). Structure and in vitro synthesis of mRNA. The basic structure of mRNA for therapeutics development. Cap, 5′ UTR, ORF, 3′ UTR, and PolyA are the five structural components of mRNA. (**B**). In vitro Synthesis of Modified mRNA. Mostly, vector DNA is used to construct the mRNA sequence. RNA polymerase uses the linearized DNA template containing either T7 or T3 or another promoter sequence for in vitro transcription (IVT) of mRNA. In some instances, caping and poly A addition can be done after the IVT process. However, some advanced kits are engineered to do the capping and poly A during the IVT process as a single step. More details are reviewed elsewhere [1].

**Figure 5 vaccines-11-00658-f005:**
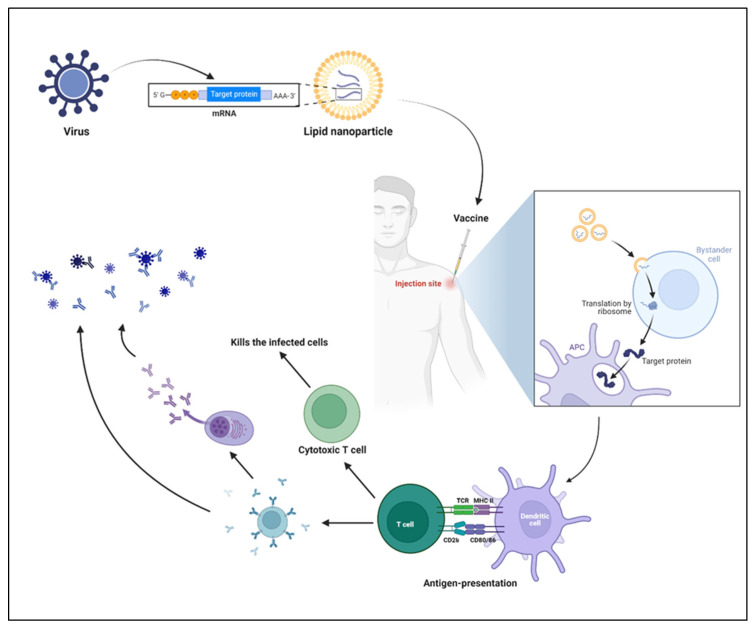
Immune response by mRNA vaccines. LNPs are prepared by encapsulating mRNA, which encodes the viral protein of interest. Upon injection of vaccines, muscular cells take up the LNPs following the release of mRNA into the cytosol and translation of target protein with the help of host machinery. In parallel, the danger associated signals produced by the LNPs recruit the innate immune cells, including neutrophils, monocytes, macrophages, dendritic cells, and others. The antigen-presenting cells (APC) process and present the antigen to the T cells, which further polarizes into effector T cells and helps in B cell-mediated responses. The cytotoxic T cells produced upon activation kill the infected cells, and antibodies (produced by B cells or plasma cells) neutralize the virus.

**Figure 6 vaccines-11-00658-f006:**
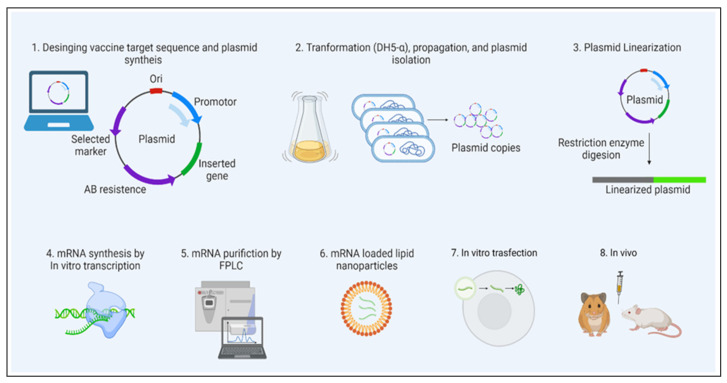
mRNA vaccine development. The figure illustrates the sequential steps involved in the mRNA vaccine development, from design to preclinical studies. Prior to going on to the next step, quality evaluation is required at each step.

## Data Availability

Not applicable.

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
