# Peer review of "Recent Advances in the Lipid Nanoparticle-Mediated Delivery of mRNA Vaccines"

_vaccines, 2023, doi:10.3390/vaccines11030658_

Round 1

Reviewer 1 Report

The review entitled “Recent Advances in the Lipid Nanoparticle-Mediated Delivery of mRNA Vaccines” reports the development of vaccines based on nanoparticles made of different classes of lipids, aimed to tackle COVID-19. This topic is extremely important and current. The authors discussed the different type of lipids forming the nanoparticles and complexing mRNA. The paper is very clear, well-structured and illustrated.

It is highly appropriate for the journal Vaccines.

I have only minor suggestions to improve the quality of the images and tables.

In addition, the authors should check the English of some sentences, like “we present recent advances and insights in the design of advanced LNPs…”; there is a repetition: advances/advanced

LNPs should be already abbreviated at line 20.

Figure 1 and 2 are placed before the text. They should be moved after they are recalled.

The chemical structures on all figures and the text into these figures are very small and difficult to read. There is enough space to increase the size of the characters of the text and the structures, including the font size of the atoms.

The names of the lipid in the tables should be in the same page of the structure.

When a name I used as a complement of specification should not have a s when plural: e.g., LNPs design should be LNP design.

This sentence is not clear: “The e 2 shows the most successful mRNA vaccines.”

Author Response

In addition, the authors should check the English of some sentences, like “we present recent advances and insights in the design of advanced LNPs…”; there is a repetition: advances/advanced.

Response: We thank the Referee for their suggestions. We corrected the grammatical errors throughout the manuscript and removed the repetitions.

  1. LNPs should already be abbreviated at line 20.

Response: Corrected the abbreviations.

  1. Figure 1 and 2 are placed before the text. They should be moved after they are recalled.

Response: Thank you once again the Referee for this suggestion; we have reorganized Figures 1-2, accordingly.

  1. The chemical structures on all figures and the text in these figures are very small and difficult to read. There is enough space to increase the size of the characters of the text and the structures, including the font size of the atoms.

Response: Once again we thank the Referee for this critical observation. After considering this, all the chemical structures in the Figures and Tables are modified including the text in the Figures. Please find the modified Figures and Tables in the Main Manuscript.

The names of the lipid in the tables should be in the same page of the structure.

  1. When a name I used as a complement of specification should not have a s when plural: e.g., LNPs design should be LNP design.

Response: We corrected these grammatical errors in the revised manuscript.

  1. This sentence is not clear: “The e 2 shows the most successful mRNA vaccines.”

Response: We corrected this typo in the revised manuscript.

Reviewer 2 Report

Please change the red color by black color in all tables

Regards

Author Response

Please change the red color by black color in all tables.

Response: We thank the Referee, now the red color in the Tables has been changed to black color.

Reviewer 3 Report

This is a very well structured and detailed review article about RNA - supported therapies highlighted by excellent figures.

The components necessary for successful application in vivo are well described.

Hence, the article is “result and effect” driven also summarized in the “unanswered questions”. In my view basic questions have not been addressed as “unanswered questions”.

-       I still miss data on the basic pharmacology of the different components. Where are the data to explain that the mRNA used against COVID-19 virus can be found for weeks in humans and even more surprisingly in milk of lactating mothers? In general, introducing any pharmacological drug, the raise and elimination/brake down in vivo of the drug must be determined. Do we have such data for mRNA related therapies?

-       Is there a connection between the components or the immune response against a given antigen like COVID-19 RBD and peri- or endocarditis in young very healthy men and less so women?

-       The authors describe formulations of a more targeted approach to reach certain organs. Do we know the path, the molecules take to reach a given organ (Fig. 3)?

-       Most important, long-term effects have not been addressed to my knowledge. The components used can most likely be incorporated into fat. Do we know a possible brake down of these products used for mRNA delivery?

Author Response

This is a very well-structured and detailed review article about RNA - supported therapies highlighted by excellent figures.

The components necessary for successful application in vivo are well described.

Hence, the article is “result and effect” driven also summarized in the “unanswered questions”. In my view basic questions have not been addressed as “unanswered questions”.

  1. I still miss data on the basic pharmacology of the different components. Where are the data to explain that the mRNA used against COVID-19 virus can be found for weeks in humans and even more surprisingly in milk of lactating mothers? In general, introducing any pharmacological drug, the raise and elimination/brake down in vivo of the drug must be determined. Do we have such data for mRNA related therapies?

Response: Thank you for the suggestion. The pharmacology of mRNA components is still not explored much, however, the mRNA optimization to improve the pharmacological aspects are listed in the revised manuscript.

We agree with the reviewer, on the biodistribution and pharmacokinetics of the mRNA. We have found the following studies, which explain the detection of mRNA in human blood (https://doi.org/10.3390/biomedicines10071538) and breast milk (https://doi.org/10.1001/jamapediatrics.2022.3581). The studies confirmed that the persistence of mRNA can be detected up to 15 days and 45 hr of post-vaccination in human blood and milk, respectively. We have used this statement in the unanswered questions in the revised manuscript (please find the highlighted text in blue color for the same). 

  1. Is there a connection between the components or the immune response against a given antigen like COVID-19 RBD and peri- or endocarditis in young very healthy men and less so women?

Response: For the movement, the present literature is not sufficient to comment on these aspects of an adverse event. However, the third and fourth points in the list of unanswered questions partially answer this question.

  1. The authors describe formulations of a more targeted approach to reach certain organs. Do we know the path, the molecules take to reach a given organ (Fig. 3)?

Response: The authors thank the Referee for their opinion. According to the previous work (Wang et al., Nat Protoc 2023, 18, 265–291, https://doi.org/10.1038/s41596-022-00755-x), authors investigated and revealed selective organ targeting (SORT) lipid nanoparticles that are methodically tailored to transport biologics exclusively to specified organs.

According to their experimental results, negatively charged SORT lipids allow for explicit delivery to the spleen. As the cationic lipid DOTAP molar proportion was increased, the luciferase protein expression moved gradually from the liver to the spleen, and subsequently to the lung, indicating a clear and precise organ-specific delivery trend with a threshold that permitted exclusive lung delivery. These findings suggest that the chemistry and proportion of SORT molecules can be adjusted for tissue-specific distribution by intravenous injection. Nevertheless, the authors did not explain the path or how to SORT nanoparticles facilitate tissue targeting phenomenon with the mechanism.

Due to copyright permission issues, we have removed the adapted Figure 3 from the revised version (please find the new Figure in the place of the old Figure 3)

Revised Figure 3. Shows the addition of a SORT lipid molecule to typical four-component LNPs alters the in vivo delivery profile of the resultant five-component SORT LNPs, allowing for tissue-specific distribution of mRNA to the liver, lungs, and spleen of mice following IV injections.

  1. Most important, long-term effects have not been addressed to my knowledge. The components used can most likely be incorporated into fat. Do we know a possible brake down of these products used for mRNA delivery?

Response:  The authors thank Referee for pointing out at this issue. Though there are minimal reports available we try to answer this comment as per our perception.

Most of the lipids used in the formulation of the lipid nanoparticles for delivering the mRNA are FDA approved and biodegradable. After releasing mRNA into the cytosolic part of the cell, most of the lipids will get involved in any of the available metabolic pathways to get cleared from the body. In designing the lipid nanoparticles for mRNA delivery both biodegradability and multifunctionality should be considered. In general, biodegradable lipids facilitate the fast elimination of lipid nanoparticles from plasma and tissues, improving their safety and tolerability. Notably, biodegradable lipids are part of the mRNA-1273 and BNT162b2 COVID-19 mRNA vaccines.